# Multimodal Emotion Recognition in Conversation Based on Hypergraphs

**Jiaze Li** **, Hongyan Mei \*, Liyun Jia and Xing Zhang**

College of Electronic and Information Engineering, Liaoning University of Technology, Jinzhou 121000, China; 219807009@stu.lnut.edu.cn (J.L.); 219915026@stu.lnut.edu.cn (L.J.); zhang_xing@lnut.edu.cn (X.Z.)

\* Correspondence: liaoning_mhy@lnut.edu.cn; Tel.: +86-189-0416-2777

**Abstract:** In recent years, sentiment analysis in conversation has garnered increasing attention due to its widespread applications in areas such as social media analytics, sentiment mining, and electronic healthcare. Existing research primarily focuses on sequence learning and graph-based approaches, yet they overlook the high-order interactions between different modalities and the long-term dependencies within each modality. To address these problems, this paper proposes a novel hypergraph-based method for multimodal emotion recognition in conversation (MER-HGraph). MER-HGraph extracts features from three modalities: acoustic, text, and visual. It treats each modality utterance in a conversation as a node and constructs intra-modal hypergraphs (Intra-HGraph) and inter-modal hypergraphs (Inter-HGraph) using hyperedges. The hypergraphs are then updated using hypergraph convolutional networks. Additionally, to mitigate noise in acoustic data and mitigate the impact of fixed time scales, we introduce a dynamic time window module to capture local-global information from acoustic signals. Extensive experiments on the IEMOCAP and MELD datasets demonstrate that MER-HGraph outperforms existing models in multimodal emotion recognition tasks, leveraging high-order information from multimodal data to enhance recognition capabilities.

**Keywords:** emotion recognition; multi-modal; hypergraph; hypergraph attention mechanism

## 1. Introduction

Language is a means of expressing and communicating emotions and accurately perceiving the emotions of others is a crucial factor in effective interpersonal communication. In conversation, humans transmit information from the speaker's brain to the listener's brain through speech. Through verbal communication, speakers not only translate their thoughts into linguistic information but also engage in the exchange and transmission of information. The task of Emotion Recognition in Conversation (ERC) aims to capture the emotional states of users in conversation, and it plays a significant role in various domains such as conversational agents, sentiment analysis, and electronic healthcare services.

For machines to effectively communicate with humans through emotions, they must possess sufficient capabilities for emotion analysis and judgment. Key factors of the ERC task include emotional stimuli (acoustic, text, visual), data collection (EEG recordings, MRI scans, facial expressions), and the ability of models to extract rich semantic features from conversation [1,2]. Traditional emotion analysis tasks employ single-modal feature extraction methods, meaning they recognize emotions from only one aspect such as acoustic, text, or video. Due to the diverse sources of emotional fluctuations, using a single modality can lead to misidentification issues, resulting in lower accuracy. Moreover, human cognitive levels are directly related to how emotions are expressed. Therefore, relying solely on a single modality makes it challenging to accurately determine emotional states. In recent years, multimodal machine learning has gained popularity as it helps compensate for the limitations of single modality in reflecting real-world situations in certain cases. Effectively modeling the interaction between utterances in conversation and enhancing the

semantic relevance of emotional information is of paramount significance for improving the performance of multimodal ERC tasks.

Currently, most multimodal ERC methods rely on Recurrent Neural Networks (RNNs) to extract sequential feature information from conversations. However, RNN-based approaches primarily propagate context and sequential information within the conversation. They simply concatenate single-modal feature information, ignoring the interaction between different modalities and the semantic relevance of the conversation context. This limitation hampers the effectiveness of multimodal ERC tasks. Since Kipf and Welling [3] introduced Graph Convolutional Networks (GCN), GCN has found wide application in various fields, such as natural language processing, computer vision, and recommendation systems. GCN, with its powerful relationship modeling capabilities, effortlessly captures long-distance contextual information in multimodal ERC tasks, modeling interactions within modalities and between different modalities. However, existing GCN-based models employ a one-to-one mapping between data, which becomes more complex when dealing with multimodal data due to the need to model data correlations.

To solve these problems, this paper proposes a multimodal ERC based on hypergraph (MER-HGraph). Firstly, acoustic, video, and text features are extracted from the conversation. Considering that acoustic data are susceptible to noise and fixed time scales, a dynamic time window is designed to process acoustic features using a Transformer model and attention mechanism. Then, the utterances from the three modalities are treated as nodes, and hypergraph convolution operations are applied to capture data correlations in the representation learning process. By constructing separate intra-modality and inter-modality hypergraphs, the modeling of modal data becomes more flexible, and it effectively facilitates interactions within the current conversation as well as between modalities. The main contributions of this paper can be summarized as follows:

- The MER-HGraph model, a multimodal conversation emotion analysis approach based on hypergraphs, is introduced. Through the design of intra-modality hypergraphs and inter-modality hypergraphs, it effectively captures context dependencies within modalities and interaction relationships between different modalities. This leads to a significant improvement in the accuracy of emotion analysis.
- The use of a dynamic time window in processing extracted acoustic features involves dynamically segmenting and re-evaluating speech signal window information through an attention mechanism. This approach effectively alleviates the impact of noise and fixed time scales inherent to acoustic signals.
- Extensive experiments were conducted on two real datasets, IEMOCAP and MELD. The results indicate that the MER-HGraph model outperforms all baseline models in the task of multimodal conversation emotion analysis.

The remaining part of the article is structured as follows. Section 2 describes related work, Section 3 offers a detailed explanation of the model's architecture and the method used for constructing hypergraphs, Section 4 presents an analysis of the experimental results, along with a breakdown of experimental parameters, and Section 5 concludes the paper.

## 2. Related Work

### 2.1. Single Modal Feature Processing

Acoustic features can be broadly categorized into two types: classical handcrafted features and those based on deep learning. Classical handcrafted features involve extracting characteristics from each frame of the acoustic signal. On the other hand, deep learning-based features dynamically capture inter-frame characteristics. Schuller [4] and others classify these two types of features into aspects such as signal energy, fundamental frequency, speech quality, cepstral coefficients, and spectrogram. Tripathi et al. [5] found that Mel-frequency cepstral coefficients (MFCCs) outperform spectrogram features. When it comes to processing acoustic features, Wang et al. [6] proposed a method of handling different time frames of speech signals through LSTM and integrating two sequences for

acoustic feature processing. Lee et al. [7] introduced a parallel fusion model that extracts temporal information from spectrograms using the BERT model, and utilizes CNN for spectrogram information extraction. Ye et al. [8] proposed a time-aware bidirectional multi-scale network, which employs a time-aware module to capture speech signal features and utilizes a bidirectional structure to model long-term dependencies.

Language features represent one way to realize speech information. With the emergence of pre-trained models, Mikolov et al. [9] introduced the Word2Vec word representation method. This approach employs a simple model to learn continuous word vectors and trains the model based on distributed representations. Devlin et al. [10] proposed the BERT pre-trained model, a novel language representation model for semantic understanding. BERT is trained based on contextual representations and excels at capturing the semantic relationships of words in different contexts. In the processing of text features, Wang et al. [11] put forward an automated method for constructing a fine-grained sentiment lexicon that encompasses sentiment information. They achieved this by extending the sentiment seed lexicon using a graph propagation method. Jassim et al. [12] combined sentiment lexicons with TF-IDF weight distribution to obtain sentence vectors, resulting in a substantial improvement over conventional sentiment lexicon methods. Xu et al. [13] proposed a CNN-based sentiment classification model, training distributed word embeddings for each word using both CNN_TEXT and the Word2Vec method. All of these methods find wide applications in text sentiment recognition, with deep learning-based approaches making significant strides in handling text sentiment recognition tasks.

Visual features also reflect changes in human emotions. Yang et al. [14] employed a Convolutional Neural Network (CNN) to obtain the output of the last layer as the global emotional feature map. By coupling this emotional heat map with the original output, a local emotional representation is formed. Combining both global and local emotional feature maps yields the classification result, enhancing the complexity of feature extraction. Guo et al. [15] utilized DenseNet to extract features from images and compared it with ResNet, BERT, and BERT-ResNet. The results demonstrated that DenseNet is more adept at feature extraction from images. Considering that excessive focus on locality may neglect overall discriminative information in target regions of the image, Li et al. [16] introduced a weakly supervised Discriminative Enhancement Network strategy. This approach applies emotional maps and discriminative enhancement maps to features, then aggregates them into an emotional vector as the basis for classification. This method better utilizes both the overall and local information in emotional images, leading to an improved classification accuracy.

### 2.2. Hypergraph Neural Network

In recent years, hypergraph learning has garnered attention due to its effectiveness in modeling high-order relationships among samples. Hypergraph learning is capable of extracting features from high-order relationships, thereby reducing information loss. This progress addresses the issue of relationships between data points extending beyond pairwise interactions. Jiang et al. [17] proposed a dynamic hypergraph convolutional neural network that dynamically updates the hypergraph structure using KNN and K-Means, enhancing its ability to capture data relationships. This allows for better extraction of both global and local relationships in the data. To better apply graph learning strategies to hypergraphs, Bai et al. [18] utilized an attention mechanism to dynamically update the hyperedge weights in the hypergraph. This not only addresses the oversmoothing issue in deep hypergraph convolution but also significantly enhances the representational capacity of the hypergraph by incorporating attention mechanisms.

There are also a few studies that integrate hypergraph learning with prediction tasks. For instance, Ding et al. [19] proposed learning two types of project embeddings based on hypergraph convolutional networks and gated recurrent units. They flexibly combined these two embeddings using an attention mechanism to obtain conversation representations. Xia et al. [20] introduced a graph convolutional network based on hypergraphs and

line graphs. They maximized the interaction between conversation representations learned by the two networks and integrated it into the network's training through self-supervised learning to enhance recommendation tasks. Ren et al. [21] treated conversation as hyperedges, merging users' repetitive behaviors within these hyperedges to form a hypergraph. This not only expresses complex relationships between unique items but also captures relationships between repetitive behaviors. In studies on other tasks, it has been observed that hypergraph neural networks are better at capturing high-order relationships within conversation, leading to improved predictive performance.

*2.3. Multimodal Emotion Recognition in Conversation*

In multimodal ERC tasks, many studies adopt sequence modeling methods to model the dependencies within each modality. For example, DialogueRNN [22] proposes the use of three GRUs to model speaker information, contextual information from the preceding dialogue, and emotional information. The global GRU and party GRU are employed to compute and update the global contextual state and the participant's state, while the emotion GRU calculates the emotional representation of the current dialogue content. AF-CAN [23] utilizes a context-aware recurrent neural network to simulate interactions and dependencies between speakers. It employs bidirectional GRU network units to capture past and future feature information. BiERU [24] extracts features from the conversation using long short-term memory units and one-dimensional convolutional neural networks. It designs a generalized neural tensor block and a dual-channel feature extractor to obtain contextual information and emotional features. However, sequence modeling methods tend to focus attention on dependencies within each modality, thereby neglecting complementary information between different modalities. This limitation restricts the performance of multimodal conversation emotion analysis. With the growing popularity of Graph Convolutional Networks (GCNs) in solving various graph-based problems, including prediction tasks and recommendation systems, DialogueGCN [25] employs a relational GCN to describe the dependencies between speakers. In the graph, nodes represent individual utterances, and edges between utterances represent dependencies between speakers and their relative positions in the conversation. RGCN [26] designs a residual convolutional neural network, generating a complex contextual feature for each individual utterance using an internal feature extractor based on ResNet. MMGCN [27] introduces a spectral domain graph convolutional network to encode multimodal contextual information, capturing speech-level contextual dependencies across multiple modalities. DSAGCN [28] proposes a conversation emotion analysis model that combines dependency parsing with GCN. It inputs feature vectors from three modalities into a Bi-LSTM, and then utilizes attention mechanisms and GCN for emotion classification. GraphMFT [29] suggests constructing three graphs (V-A graph, V-T graph, and A-T graph) for each conversation and extracts intra-modal and inter-modal interaction relationships using an improved Graph Attention Network (GAT). These methods either fail to capture interactions between different modalities or overlook the heterogeneity of multimodal data. Moreover, existing multimodal ERC tasks mostly employ GCN to model interactions between different modalities. However, GCN simplifies the relationships between feature data to binary relations, resulting in the loss of many high-order associations present in the original data. Thus, the MER-HGraph is proposed. The Laplacian matrix of the hypergraph extends the node neighborhoods, enabling it to aggregate richer high-order information, and consequently more accurately model multi-order associations. The information of the multimodal conversation sentiment analysis models is shown in Table 1.

**Table 1.** Multimodal conversation sentiment analysis models.

| Model | Author Contribution | Experiment Result |
|---|---|---|
| DialogueRNN | Uses three GRUs to model speaker information, contextual information from the preceding dialogue, and emotional information. | Datasets: IEMOCAP Acc/wa-F1: 63.40/62.75 |
| AF-CAN | Utilizes a context-aware recurrent neural network to simulate interactions and dependencies between speakers. | Datasets: IEMOCAP Acc/wa-F1: 64.60/63.70 |
| BiERU | Extracts features from the conversation using long short-term memory units and one-dimensional convolutional neural networks. | Datasets: IEMOCAP/MELD Acc/wa-F1: 66.09/64.59 Average: 60.9 |
| DialogueGCN | Employs a relational GCN to describe the dependencies between speakers in the graph. | Datasets: IEMOCAP/MELD Acc/wa-F1: 65.54/65.04 Acc/wa-F1: 58.62/56.36 |
| RGCN | Generates a complex contextual feature for each individual utterance using an internal feature extractor based on ResNet. | Datasets: IEMOCAP/MELD Average: 65.08 Average: 55.98 |
| MMGCN | Designs a spectral domain graph convolutional network to encode multimodal con-textual information. | Datasets: IEMOCAP/MELD Average: 66.22 Average: 58.65 |
| DSAGCN | Proposes a conversation emotion analysis model that combines dependency parsing with GCN. | Datasets: IEMOCAP/MELD Acc/wa-F1: 63.50/61.70 Acc/wa-F1: 60.90/58.70 |
| GraphMFT | Constructs three graphs (V-A graph, V-T graph, and A-T graph). | Datasets: IEMOCAP/MELD Acc/wa-F1: 67.9/68.07 Acc/wa-F1: 61.30/58.37 |

## 3. Model and Methods

In the task of multimodal emotion analysis in conversation, this paper proposes the MER-HGraph model as shown in Figure 1. The specific model comprises single-modality feature extraction, a multimodal conversation hypergraph network, and an emotion prediction layer. The multimodal conversation hypergraph network encompasses speaker embedding, hypergraph construction, and hypergraph convolutional networks.

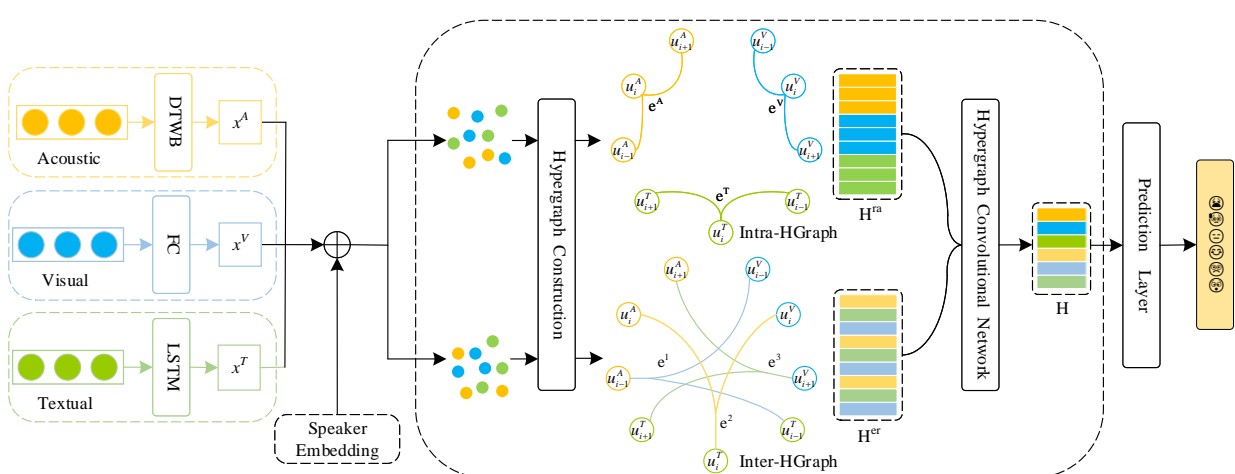

**Figure 1.** Multimodal emotion recognition in conversation based on hypergraphs.

### 3.1. Problem Definition

In multimodal ERC tasks, each conversation consists of a total of $n$ utterances, which can be defined as $\{u_1, u_2, \ldots, u_n\}$. Each utterance $(u_i^V, u_i^A, u_i^T)$ is represented in three

modalities: V for visual, A for acoustic, and T for text. The objective of the multimodal ERC task is to learn to predict the corresponding emotion of $u_i$ by leveraging both the dependencies within each modality and the interactions across modalities.

### 3.2. Single Modal Feature Extraction

We employ DenseNet, OpenSMILE, and TextCNN to respectively extract features from the visual, acoustic, and text modalities. Considering that acoustic signals are susceptible to noise interference and face issues related to fixed time scales, we designed a Dynamic Temporary Window (DTWB) to process the extracted acoustic features, as illustrated in Figure 2.

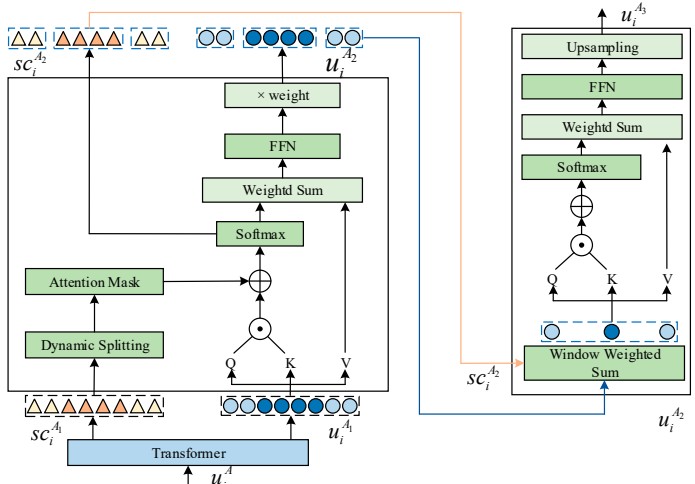

**Figure 2.** Dynamic temporary window block.

First, the local dynamic window processes the acoustic signal $u_i^A$ through a Transformer model to obtain $u_i^{A_1}$, and computes scores $sc_i^{A_1} \in \mathbb{R}^T$. These scores are then used to partition the time sequence into several strong or weak emotion windows based on a threshold set at the median of the scores. To handle acoustic signals in batches, window segmentation is implemented using an attention mask mechanism. The calculations are as follows:

$$M_{ij} \begin{cases} 0, & (b_{w_k} \le i \le e_{w_k}, b_{w_k} \le j \le e_{w_k}) \\ -\infty, & \text{others} \end{cases} \tag{1}$$

$$u_i^{A_2} = FFN(\text{Softmax}(\frac{Q_i^{A_1}\left(K_i^{A_1}\right)^{\mathrm{T}}}{\sqrt{d_h}} + M)V_i^{A_1}) \tag{2}$$

where $M_{ij}$ denotes the value of attention mask $M \in \mathbb{R}^{T \times T}$ at the $i$-th row and $j$-th column; $k = 1, \dots, n$, $b_{w_k}$, and $e_{w_k}$ are the start and end indices of the $k$-th window's row and column, respectively; $Q_i^{A_1}, K_i^{A_1}, V_i^{A_1}$ denote the projection mapping of $u_i^{A_1}$; the output is defined as $u_i^{A_2} \in \mathbb{R}^{T \times D}$. The global dynamic window module reevaluates the importance between windows by taking input $u_i^{A_2}$ and calculating scores $sc_i^{A_2}$. In this process, each window is first used to generate a new token through weighted summation, as calculated by the following formulas:

$$wt_k = \sum_{p=b_{w_k}}^{e_{w_k}} sc_i^{A_2} \times u_i^{A_2} \tag{3}$$

$$u_i^{A_3} = FFN(\text{Softmax}(\frac{Q_i^{wt}\left(K_i^{wt}\right)^{\mathrm{T}}}{\sqrt{d_h}} + M)V_i^{wt}) \tag{4}$$

By duplicating the window token upsampling for each window to match their respective lengths, and concatenating them together generates the sequence $u_i^{A3} \in \mathbb{R}^{T \times D}$. Finally, we fuse features from modality $u_i^{A2}$ and modality $u_i^{A3}$ to obtain the acoustic features. We employ fully connected networks to process the visual modality features, enhancing their representational power. For the text modality, we utilize a bidirectional LSTM network to extract contextual information from the utterances. The computation process for single modality encoding is as follows:

$$
\begin{aligned}
x_i^A &= u_i^{A2} \oplus u_i^{A3} \\
x_i^T &= \overleftrightarrow{LSTM}_e(u_i^T, h_{i-1}^T, h_{i+1}^T) \\
x_i^V &= W_e^V u_i^V + b_i^V
\end{aligned}
\tag{5}
$$

where $u_i^A, u_i^T, u_i^V$ denote the input for acoustic, text, and visual modalities, and $x_i^A, x_i^T, x_i^V$ respectively denote the encoded outputs for acoustic, text, and visual modalities.

*3.3. Multimodal Conversation Hypergraph Network*
3.3.1. Speaker Embedding

Since there are a large number of participants in each conversation, speaker information plays a crucial role in multimodal ERC tasks. To fully leverage this information, we use a one-hot vector $s_i$ to represent speaker information. It is integrated with multimodal features before hypergraph construction to obtain a new fused representation of utterances with integrated speaker information. The speaker encoding can be represented as:

$$
S_i = W_s s_i + b_i^s
\tag{6}
$$

3.3.2. Hypergraph Learning

Due to the powerful expressive capabilities of hypergraph neural networks (HGNNs) in representation learning, we adopt an HGNN to describe relationships within the conversation. Let $G = (U, E, H)$ represent a hypergraph, where the vertex set $u_i \in U$ and hyperedge set $\varepsilon \in E$ contain $N$ unique nodes and $M$ hyperedges, respectively, and $H \in R^{N \times M}$ is the incidence matrix between hyperedges and vertices, defined as:

$$
h(u, \varepsilon) = \begin{cases} 1, & u \in \varepsilon \\ 0, & u \notin \varepsilon \end{cases}
\tag{7}
$$

We treat each utterance in the conversation as a vertex, forming the set $U$. All conversations form a hyperedge $E$. By sharing vertices, we connect hyperedges to construct the hypergraph for multimodal conversation emotion analysis. For the hypergraph $G$, $D \in R^{N \times N}$ is the diagonal matrix of vertex degrees, and $B \in R^{M \times M}$ is the diagonal matrix of hyperedge degrees, defined as:

$$
\begin{aligned}
D_{ii} &= \sum_{\varepsilon=1}^{N} W_{\varepsilon\varepsilon} H_{i\varepsilon} \\
B_{\varepsilon\varepsilon} &= \sum_{i=1}^{N} H_{i\varepsilon}
\end{aligned}
\tag{8}
$$

To address the node classification problem on the hypergraph, where node labels should be smooth across the hypergraph's structure, a regularization framework is employed for hypergraph classification. The calculation is as follows:

$$
\underset{f}{\arg\min} = \{\Omega(f) + \lambda R_{emp}(f)\}
\tag{9}
$$

where $\Omega(f)$ is the hypergraph regularization term; $R_{emp}(f)$ is the supervised empirical loss; $f(\cdot)$ is the classification function; and $\lambda$ is a non-negative parameter.

### 3.3.3. Hypergraph Construction

In order to capture the dependencies between utterances in the conversation and the interactions between different modalities, we use the extracted features from three modalities as input to construct intra-modal hypergraphs (Intra-HGraph) and inter-modal hypergraphs (Inter-HGraph) for each conversation. (1) Intra-HGraph refers to the contextual dependency relationships between utterances in a conversation. In the conversation, let $u^A, u^T, u^V$ represent a node, where each node denotes an utterance in its modality. There are $3 \times m$ nodes, where $m$ is the number of utterances in the current modality in the conversation. Three types of hyperedges, denoted as $(\varepsilon^A, \varepsilon^T, \varepsilon^V)$, are created within each modality. Each node is connected to past $P$ and future $F$ context nodes within the current modality. (2) Inter-HGraph refers to the interaction relationships between different modalities within the same utterance. The nodes in the Inter-HGraph are the same as those in the Intra-HGraph. We connect each node to nodes from the same utterance but belonging to different modalities, constructing inter-modality hyperedges $(\varepsilon^1, \varepsilon^2 \ldots, \varepsilon^n)$.

Considering that different adjacent nodes may have varying impacts on the current utterance node and different modalities of the same node may interact differently, we assign weights $W_{\varepsilon\varepsilon}$ to each hyperedge, and these weights form the diagonal matrix $W \in R^{M \times M}$ of hyperedge weights for this hypergraph. Additionally, we construct association matrices $H^{ra}$ and $H^{er}$ between nodes and hyperedges for both the Intra-Hgraph and Inter-HGraph.

### 3.3.4. Hypergraph Convolution

The hypergraph convolution operation efficiently utilizes higher-order relationships and local clustering structures to achieve effective information propagation between vertices. The process of multimodal conversation hypergraph convolution is illustrated in Figure 3. This process can be divided into two stages: (1) information aggregation from vertices to hyperedges; (2) information aggregation from hyperedges to vertices. Specifically, the information from each vertex is aggregated into the corresponding hyperedge, resulting in a representation for each hyperedge. Then, the hyperedges connected to each vertex are located, and their information is aggregated into the vertex, generating a representation for each vertex.

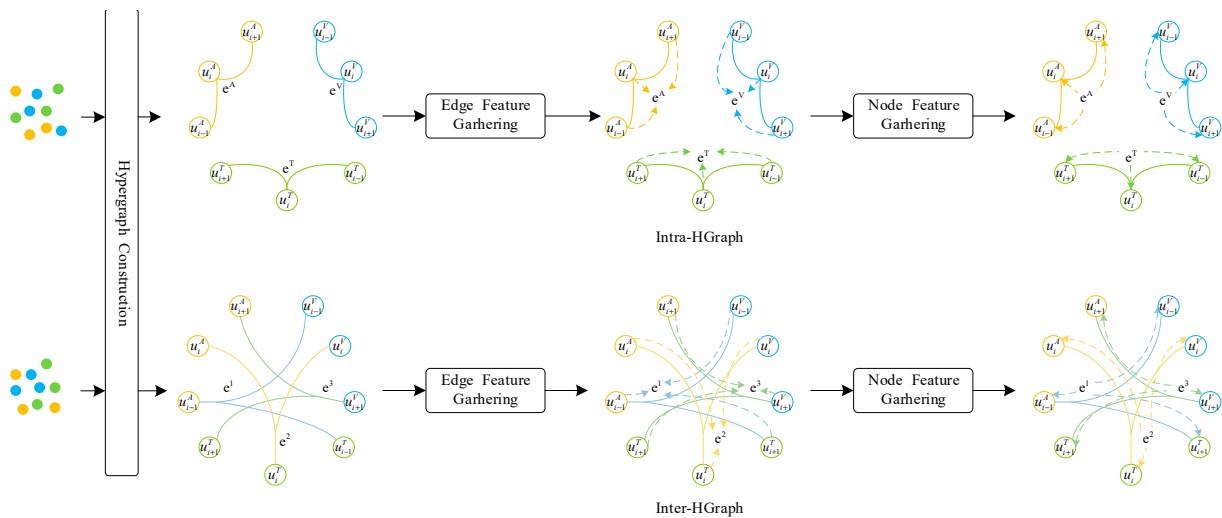

**Figure 3.** Multimodal hypergraph convolution process.

In the Intra-HGraph, we aggregate the information of vertex $u^A_{i-1}, u^A_i, u^A_{i+1}$ to edge $e^A$, the information of vertex $u^T_{i-1}, u^T_i, u^T_{i+1}$ to edge $e^T$, and the information of vertex $u^V_{i-1}, u^V_i, u^V_{i+1}$ to edge $e^V$. In the Inter-HGraph, we aggregate the information of vertex $u^A_{i-1}, u^T_{i-1}, u^V_{i-1}$ to edge $e^1$, the information of vertex $u^A_i, u^T_i, u^V_i$ to edge $e^2$, and the information of vertex $u^A_{i+1}, u^T_{i+1}, u^V_{i+1}$ to edge $e^3$. Through this process, MER-HGraph obtains representations for all vertices in both intra-modal and cross-modal aspects, further enhanc-

ing the learning of conversation representations. We define the hypergraph convolution as follows:

$$x_i^{(l+1)} = \sum_{j=1}^{N} \sum_{\varepsilon=1}^{M} H_{i\varepsilon} H_{j\varepsilon} W_{\varepsilon\varepsilon} x_j^{(l)} \tag{10}$$

where $x_i^{(l+1)}$ denotes the $(l+1)$-th layer and $t$-th node. Each $W_{jj}$ for $j$ is set to 1. We do not employ non-linear activation functions and convolutional filter parameter matrices. For $W_{\varepsilon\varepsilon}$, we assign the same weight of 1 to each hyperedge. The row normalization matrix form of Equation (11) is given by:

$$X_h^{(l+1)} = D^{-1} H W B^{-1} H^T X_h^{(l)} \tag{11}$$

Hypergraph convolution can be viewed as a two-stage evolution of feature transformation on the hypergraph structure, performing a "node-hyperedge-node" transformation. By concatenating the two incidence matrices, $H^{ra}$ and $H^{er}$, for Intra-HGraph and Inter-HGraph, we obtain the final incidence matrix $H$. The multiplication operation $H^T X_h^{(l)}$ defines the aggregation of information from nodes to hyperedges, followed by pre-multiplication by $H$ to aggregate information from hyperedges back to nodes.

### 3.4. Multimodal Emotion Prediction in Conversation

We use the obtained incidence matrix $h_i \in H$ as input for a fully connected network for emotion prediction, and its computation formula is as follows:

$$z_i = \text{ReLU}(wh_i + b) \tag{12}$$

$$p_i = \text{Softmax}(w'z_i + b') \tag{13}$$

$$\hat{y}_i = \underset{k}{\text{argmax}}(p_i[\text{k}]) \tag{14}$$

where $h_i \in H$ denotes the final feature vector of the $i$-th utterance $u_i$; ReLU denotes the non-linear activation function; $p_i$ denotes for the predicted emotion probability distribution of $u_i$; $\hat{y}_i$ denotes the predicted emotion, and $w, w', b, b'$ denotes trainable parameters. We employ the cross-entropy loss function as the objective function for training, which is computed as follows:

$$Loss = -\frac{1}{\sum_{t=0}^{N-1} n(t)} \sum_{i=0}^{N-1} \sum_{j=0}^{n(i)-1} y_{ij} \log p_{ij} + \lambda ||W_{ls}||_2 \tag{15}$$

where $N$ is the total number of conversations in the dataset, $n(i)$ is the number of utterances in the $i$-th conversation; $y_{ij}$ denotes the true emotion of the $j$-th utterance in the $i$-th conversation; $p_{ij}$ denotes the predicted emotion probability distribution of the $j$-th utterance in the $i$-th conversation; $\lambda$ denotes the L2-regularization weight; and $W_{ls}$ denotes the set of learnable parameters.

## 4. Experiment

### 4.1. Experimental Environment

The experimental environment was based on the Ubuntu 20.04 operating system, equipped with an Intel i7-11800H CPU, NVIDIA GeForce RTX 3060 GPU, and 12 GB of memory. The development environment utilized a TensorFlow deep learning framework, Python 3.8, and CUDA 14.1. A cross-entropy criterion was employed as the objective function for model training, and the Adam optimization algorithm was used to update model parameters.

*4.2. Datasets*

In this study, extensive experiments were conducted on two widely used public datasets, MELD and IEMOCAP. Both datasets are multimodal, containing modalities of acoustic, text, and vision. The statistical summary of these two datasets is presented in Table 2.

**Table 2.** Data distribution of IEMOCAP and MELD.

| Dataset | Conversations | | Utterances | |
|---------|:-----------:|:----:|:-----------:|:----:|
| | **Train + Val** | **Test** | **Train + Val** | **Test** |
| IEMOCAP | 120 | 31 | 5810 | 1623 |
| MELD | 1153 | 280 | 11098 | 2610 |

The IEMOCAP dataset consists of recordings of ten actors engaged in dyadic interactions, organized into five conversations, each involving one male and one female participant. This dataset provides three modalities: acoustic, text, and visual, with 7433 (5810 + 1623) text utterances and approximately 12 h of acoustic and video. Each utterance can be labeled with one of six different emotion categories: happy, sad, neutral, angry, excited, and frustrated.

The MELD dataset is derived from the EmotionLines dataset and features multiple speakers in conversation. It consists of 1153 (1039 + 114), and 280 conversations for training, validation, and testing, respectively, all from the TV series "Friends". Each conversation is labeled with one of the following emotion categories: anger, disgust, sadness, joy, surprise, fear, and neutral.

*4.3. Experimental Result and Analysis*

4.3.1. Evaluation Metrics

To evaluate the performance of the model, accuracy and weighted average F1 score are used as evaluation metrics. Accuracy measures the correctness of the model's predictions, while the weighted average F1 score considers both precision and recall. Its calculation formula is as follows:

$$
\begin{aligned}
Accuracy &= \frac{TP+TN}{TP+TN+FP+FN} \\
Precision &= \frac{TP}{TP+FP} \\
Recall &= \frac{TP}{TP+FN} \\
F1 &= 2 \times \frac{Precision \times Recall}{Precision + Recall} \\
wa - F1 &= \sum_{i=1}^{N} w_i \times F1_i
\end{aligned}
\tag{16}
$$

where $TP$ represents True Positives; $TN$ represents True Negatives; $FP$ represents False Positives; $FN$ represents False Negatives.

4.3.2. Baseline Methods

To validate the effectiveness of the model, it was compared against several baseline methods:

(1) The bc-LSTM [30] is a method proposed to capture contextual features from surrounding utterances using bidirectional LSTM. However, it does not take into account the interdependence between speakers.

(2) ICON [31] utilizes two separate GRUs to model the contextual information of utterances from two speakers in the dialogue history. The current utterance serves as the query input to two distinct speaker memory networks, generating utterance representations. Another GRU connects the output of the individual speaker GRUs in the CMN, explicitly modeling the interplay between speakers.

(3) DialogueRNN [22] proposes the use of two GRUs to track the state of individual speakers and the global context within the conversation. Additionally, another GRU is employed to track the emotional states throughout the conversation. DialogueRNN can be applied to various datasets and models the relationships between speakers.

(4) DialogueGCN [25] introduces an emotion recognition method based on graph neural networks. It models the contextual information for emotion recognition by utilizing the self-dependency of speakers and the dependency between speakers, addressing the issue of context propagation that exists in current RNN-based methods.

(5) DialogueCRN [32] introduces a cognitive phase to extract and integrate emotional cues from context, successfully utilizing these cues for improved emotion state classification. Multimodal features are combined to facilitate a multimodal setting.

(6) MMGCN [27] simultaneously learns multimodal and long-term contextual dependencies through deep graph convolutional neural networks. Speaker information is mapped to a one-hot vector to model dependencies between speakers.

(7) COGMEN [33] proposes a multimodal emotion architecture with a contextual graph neural network. It leverages both local information (interactions between speakers) and global information (context), modeling complex dependencies in the dialogue using Graph Convolutional Networks (GCN).

(8) GraphMFT [29] treats each data object in the conversation as a node, with intra-modal and cross-modal dependencies considered as edges. It employs multiple enhanced graph attention networks to capture both intra-modal contextual information and inter-modal complementary information.

The experimental results are shown in Table 3. We compared the performance of our model on the test data with other commonly used methods.

**Table 3.** Experimental results of different models on IEMOCAP and MELD datasets. Evaluation metrics contain Acc, F1, and wa-F1, which denote accuracy score (%), F1 score (%), and weighted-average F1 score (%), respectively.

| Model | IEMOCAP | | | | | | | | MELD | |
|---|---|---|---|---|---|---|---|---|---|---|
| | Happy F1 | Sad F1 | Neutral F1 | Angry F1 | Excited F1 | Frustrated F1 | Acc | wa-F1 | Acc | wa-F1 |
| bc-LSTM | 32.63 | 70.34 | 51.14 | 63.44 | 67.91 | 61.06 | 59.58 | 59.10 | 59.62 | 56.80 |
| ICON | 29.91 | 64.57 | 57.38 | 63.04 | 63.42 | 60.81 | 59.09 | 58.54 | - | - |
| DialogueRNN | 33.18 | 78.80 | 59.21 | 65.28 | 71.86 | 58.91 | 63.40 | 62.75 | 60.31 | 57.66 |
| DialogueGCN | 47.10 | 80.88 | 58.71 | 66.08 | 70.97 | 61.21 | 65.54 | 65.04 | 58.62 | 56.36 |
| DialogueCRN | 51.59 | 74.54 | 62.38 | 67.25 | 73.96 | 59.97 | 65.31 | 65.34 | 59.66 | 56.76 |
| MMGCN | 45.45 | 77.53 | 61.99 | 66.67 | 72.04 | 64.12 | 65.36 | 65.71 | 59.31 | 57.82 |
| COGMEN | 51.90 | 81.72 | 68.62 | 66.03 | 75.34 | 58.21 | 67.85 | 67.62 | - | - |
| GraphMFT | 45.99 | 83.12 | 63.08 | 70.30 | 76.92 | 63.84 | 67.90 | 68.07 | 61.30 | 58.37 |
| MER-HGraph | 52.34 | 83.47 | 68.51 | 71.83 | 77.41 | 65.28 | 70.81 | 70.37 | 62.76 | 59.13 |

According to the results in Table 3, the MER-HGraph model outperforms other baseline models on the IEMOCAP dataset in terms of both Accuracy and Weighted-average F1 scores. The model achieves an accuracy of 70.81% and a weighted-average F1 score of 70.37%, which are 2.91% and 2.3% higher, respectively, compared to the best-performing baseline model. In contrast to traditional sequence modeling approaches like bc-LSTM, ICON, DialogueRNN, and DialogueCRN, which may not comprehensively exploit and utilize contextual information in the conversation and fail to leverage the interactions across modalities effectively, MER-HGraph employs HGNN to model utterances in the conversation. It captures Intra-HGraph and Inter-HGraph dependencies to better accomplish the task of multimodal conversation sentiment analysis by considering both contextual dependencies within modalities and interactions across modalities. Compared to MMGCN, COGMEN, and Graph-MFT, all three models utilize GCN to model dependencies among

speakers and contextual information. However, GCN adopts a pairwise interaction approach, which overlooks higher-order information in the conversation. MER-HGraph, on the other hand, leverages HGNN to aggregate information from each vertex to its corresponding hyperedge, obtaining hyperedge representations. It then aggregates hyperedge information back to vertices, yielding high-order information within the conversation. Additionally, MER-HGraph employs a dynamic time window to reduce the impact of noise and fixed time scales in acoustic signals. Furthermore, on the MELD dataset, MER-HGraph achieves an accuracy score and weighted-average F1 score improvement of 1.46% and 0.76%, respectively, compared to the best-performing model. Overall, the proposed MER-HGraph model effectively enhances the capability of multimodal conversation sentiment analysis by modeling contextual dependencies and cross-modal interactions using HGNN, and by introducing a dynamic time window to mitigate the impact of acoustic signal noise, as demonstrated on both the IEMOCAP and MELD datasets.

### 4.3.3. Impact of Dynamic Temporary Window Block

Considering the influence of temporal information localization on sentiment analysis performance, experiments were conducted using both fixed and dynamic time windows, as illustrated in Figure 4. Here, the horizontal axis represents the chronological order, and the vertical axis represents the importance scores over time. [lau] represents laughter, [y] denotes affirmative tone, [noise] indicates noise, and [s] denotes silence. The yellow region represents the area of interest, while the blue region indicates the area that should not be considered.

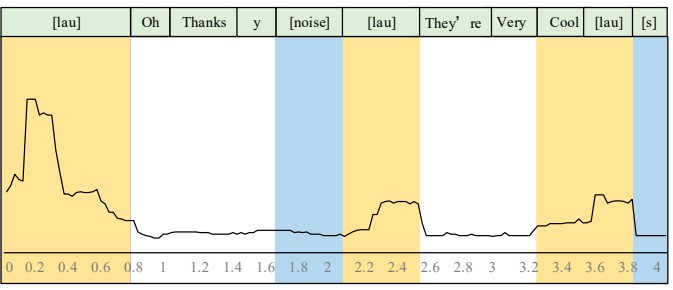

(a) Dynamic time window

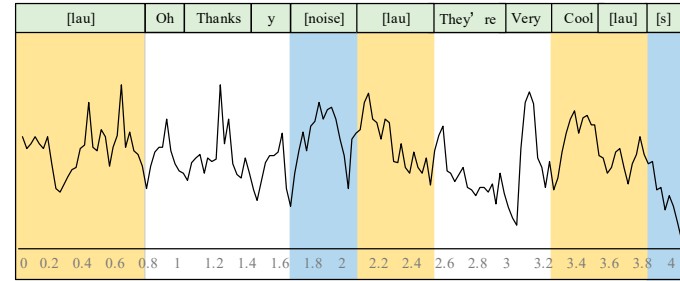

(b) Fixed time window

**Figure 4.** Impact of dynamic windows and fixed windows. The horizontal axis represents chronological order and vertical axis is of importance score.

The experimental results indicate that the dynamic temporary window module, by partitioning the signal into different lengths of strong and weak emotions locally, and assessing the interaction information between emotions globally, achieves better performance. For instance, in areas of interest such as laughter and positive semantics ("Cool"), the dynamic window provides a smoother signal processing compared to the fixed window. In areas that should not be focused on, like noise and silence, the dynamic window almost disregards the signal information, while the fixed window shows significant fluctuations in information processing. This experiment validates that the dynamic time window effectively captures both local and global signal features in acoustic.

### 4.3.4. Impact of the Number of Context Nodes

When capturing intra-modal dependencies, the current node needs to connect to past $P$ and future $F$ contextual nodes. The influence of the number of contextual nodes $(P, F)$ on MER-HGraph is considered. In the IEMOCAP dataset, it is set to $(4, 4), (8, 8), \cdots (40, 40)$, and in the MELD dataset, it is set to $(1, 1), (2, 2), \cdots (10, 10)$. The impact of different numbers of contextual nodes on the accuracy scores and weighted average F1 scores of MER-HGraph is shown in Figure 5.

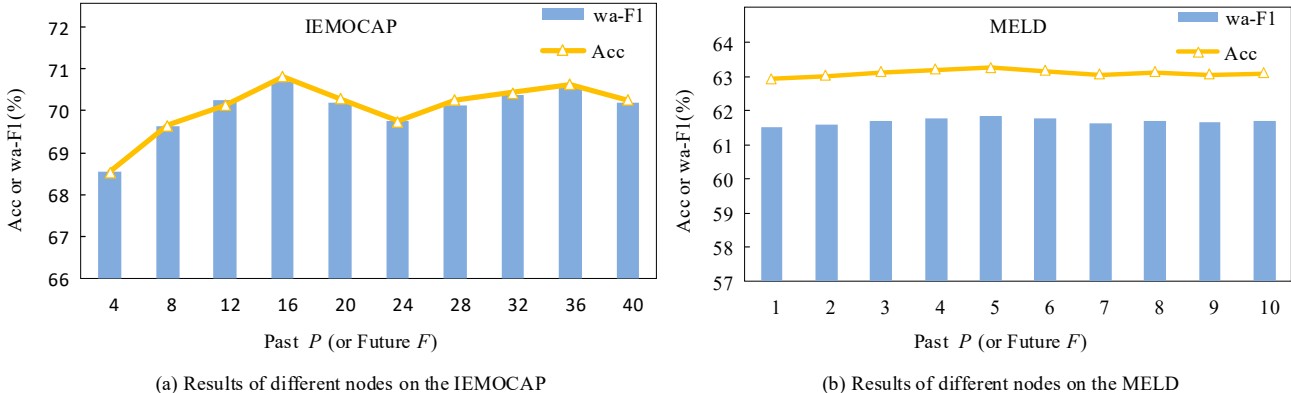

**Figure 5.** Impact of the number of context nodes.

In Figure 5a, it can be observed that the performance of MER-HGraph on the IEMO-CAP dataset increases with the increase in $P$ or $F$. When $(P, F)$ reaches the threshold of $(16, 16)$, the accuracy score and F1 score of the MER-HGraph model achieve the best performance. In Figure 5b, it can be observed that with the MELD dataset, the accuracy score and F1 score of the MER-HGraph model reach the best performance when $(P, F)$ reaches the threshold of $(5, 5)$. It is possible that the IEMOCAP dataset requires longer contextual information modeling, while in the MELD dataset, some adjacent utterances are not necessarily adjacent in actual scenarios, so there is no need for longer contextual modeling.

*4.4. Ablation Study*

The ablation experiments were conducted to further validate the roles and importance of different parts of the model. The results of the ablation experiments are shown in Table 4. Here, "-" indicates that the corresponding part was removed, while "+" indicates that the corresponding part was used. In the first case of the ablation experiments, the HGNN network was replaced with a GCN network. In the second case, the DTWB was replaced with a fully connected network. The third case represents the proposed MER-HGraph model in this paper.

**Table 4.** Ablation study for the main components in MER-HGraph on the IEMMOCAP and MELD dataset.

| HGNN | DTWB | IEMOCAP | MELD |
|---|---|---|---|
| - | + | 66.54 | 57.85 |
| + | - | 69.31 | 58.62 |
| + | + | 70.37 | 59.13 |

Based on the results in Table 4, we observe that HGNN outperforms GCN. This is attributed to the fact that GCN acquires information within modalities and interactions across multimodalities through pairwise connections. On the other hand, HGNN constructs hypergraphs where a single hyperedge can connect multiple speech nodes. Moreover, HGNN can simultaneously link acoustic, text, and visual modalities through a single hyperedge. As a result, HGNN can capture higher-order information in the data, reducing information loss and thereby improving the performance of the emotion analysis task. Additionally, the DTWB module is designed to handle audio signals, reducing the impact of noise and fixed time scales. This allows for better capturing of temporal sequences in speech signals, consequently enhancing the performance of emotion classification.

**5. Conclusions**

For the task of multimodal emotion analysis in conversation, we propose a method based on a hypergraph neural network. Unlike previous studies, we introduced an HGNN

to construct Intra-Hgraph and Inter-HGraph within conversation to capture dependencies between utterances and interactions between different modalities. Additionally, to address the issues of noise and fixed time scales in speech signals, we designed a dynamic time window to extract local and global information from the audio signals. Through this approach, MER-HGraph can acquire richer feature information, thereby enhancing the effectiveness of emotion analysis tasks. The proposed model is evaluated on the IEMOCAP and MELD datasets and compared with other baseline models, demonstrating that MER-HGraph outperforms them. Given the wide application of hypergraph neural networks in other research fields, this study introduces hypergraphs into the task of multimodal emotion analysis, and future improvements to HGNNs could further enhance the performance of multimodal conversation emotion analysis tasks.

**Author Contributions:** Conceptualization, J.L.; methodology, J.L.; software, HM; validation, J.L., H.M. and L.J.; formal analysis, L.J.; investigation, J.L.; resources, J.L.; data curation, J.L.; writing—original draft preparation, H.M.; writing-review and editing X.Z.; visualization, X.Z.; supervision, X.Z.; project administration, J.L.; funding acquisition, H.M. All authors have read and agreed to the published version of the manuscript.

**Funding:** This work is supported by the Liaoning Education Department Scientific Research Project (No. JZL202015404, No. LIKZ0625), the General project of Liaoning Provincial Department of Education (No. LJKZ0618).

**Data Availability Statement:** Experiments used publicly available datasets.

**Conflicts of Interest:** The authors declare no conflict of interest.

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
