# Peer review of "Multimodal Emotion Recognition in Conversation Based on Hypergraphs"

_electronics, doi:10.3390/electronics12224703_

Round 1
Reviewer 1 Report
Comments and Suggestions for Authors
The authors of this paper introduce a novel approach called MER-HGraph (Multimodal Emotion Recognition using Hypergraph) for sentiment analysis in conversation. They address the limitations of existing methods by incorporating high-order interactions between different modalities (acoustic, text, and visual) and considering long-term dependencies within each modality.
- The related work section lacks any results. I suggest adding results for each mentioned work and creating a summary table for them.
- The values in Table 1 are inconsistent with the dataset description at the top of the table.
- Performance calculation equations should be written.
- The paper needs a meticulous language review; there are errors, including but not limited to (to order to in the introduction, and modles in section 3:
Author Response
Thank you very much for your review of my paper. I have revised the paper as required, and the following are the answers to the questions.
The authors of this paper introduce a novel approach called MER-HGraph (Multimodal Emotion Recognition using Hypergraph) for sentiment analysis in conversation. They address the limitations of existing methods by incorporating high-order interactions between different modalities (acoustic, text, and visual) and considering long-term dependencies within each modality.
Q: The related work section lacks any results. I suggest adding results for each mentioned work and creating a summary table for them.
A: This article added a summary table of multimodal conversational emotion analysis models in the relevant work section.
Q: The values in Table 1 are inconsistent with the dataset description at the top of the table.
A: The original text did not provide an accurate description of the datasets. For the IEMOCAP dataset, it should be 7433 utterances (5810 + 1623). For the MELD dataset, it is divided into 1153 (1039 + 114) and 280 dialogues. This has now been corrected.
Q: Performance calculation equations should be written.
A: The article added content on evaluation metrics in section 4.3.1, including the performance calculation formulas.
Q: The paper needs a meticulous language review; there are errors, including but not limited to (to order to in the introduction, and modles in section 3).
A: I am very sorry that my poor English writing ability has brought some trouble to your review. At present, I have revised the overall grammar and expression of the paper.

Reviewer 2 Report
Comments and Suggestions for Authors
The submitted manuscript titled as "Multimodal Emotion Recognition in Conversation Base on Hypergraphs", has introduced a novel hypergraph based on multimodel emotion recognization (MER-Hgraph) in conversation. Each modality utterance in conversation are processed using intra-modal and inter-modal using hyper edges. However, I have some comments:
Specific Comments:
1) What are the significance of strong or weak emotion windows and why median of the scores is useful for extraction?
2) Since convolution operation is baseline for the research work, how two-stage evolution of feature transformation on the hypergraph structure is verified?
4) It is better to develop an algorithm or flowchart for the overall desiging process.
3) It is better to demonstrate textual, acuostic and visual results to enhance readabiltiy.
Author Response
Thank you very much for your review of my paper. I have revised the paper as required, and the following are the answers to the questions.
The submitted manuscript titled as "Multimodal Emotion Recognition in Conversation Base on Hypergraphs", has introduced a novel hypergraph based on multimodel emotion recognization (MER-Hgraph) in conversation. Each modality utterance in conversation are processed using intra-modal and inter-modal using hyper edges. However, I have some comments:
Q: What are the significance of strong or weak emotion windows and why median of the scores is useful for extraction?
A: The proposed strong and weak emotion windows in this paper are intended for further analysis of the emotional intensity in different time periods of the audio signal. The extraction of emotion windows helps in understanding emotional changes in the audio signal, reducing fluctuations caused by signal elements such as noise and silence. Choosing the median of the scores as the threshold is done to take into account the distribution of emotional scores. The median is a statistic unaffected by extreme values, making it a more robust choice for threshold determination, ensuring a relatively balanced partition across the overall distribution of emotional scores.
Q: Since convolution operation is baseline for the research work, how two-stage evolution of feature transformation on the hypergraph structure is verified?
A: Hypergraph convolution involves a two-stage transformation process. Firstly, the information from each vertex is aggregated into the located hyperedge to obtain an edge degree matrix. Next, the hyperedges connected to each vertex are identified, and the information from these hyperedges is aggregated into the vertices, resulting in a vertex degree matrix. Subsequently, the hypergraph adjacency matrix H is generated.
Q: It is better to demonstrate textual, acuostic and visual results to enhance readabiltiy.
A: The presentation of unimodal results enhances readability. The authors plan to contemplate and enrich experimental results in future experiments.
Q: It is better to develop an algorithm or flowchart for the overall desiging process.
A: The paper optimizes the multimodal hypergraph convolution process in Figure 3.

Reviewer 3 Report
Comments and Suggestions for Authors
Before this article, I thought that the Bert model was the best.
I always have skepticism on the topic of emotion recognition, since there are always individual characteristics in this area. This work is beyond the scope of this topic and can be applied in other areas.
Hypergraph Neural Network allow to define new level of analysis.
Part of Hypergraph Convolution very interesting. But how it will work in high-dimensional space? for example, when analyzing spectral characteristics?
Author Response
Thank you very much for your review of my paper. I have revised the paper as required, and the following are the answers to the questions.
Q: I always have skepticism on the topic of emotion recognition, since there are always individual characteristics in this area. This work is beyond the scope of this topic and can be applied in other areas.
A: The core of this paper lies in handling the intra-modal and cross-modal modeling aspects. Therefore, three different approaches were chosen based on the experiences of other models to handle the three modalities. The BERT model utilizes the encoder part of the Transformer model and employs bidirectional modeling to better capture contextual information. However, models based on BERT undergo deep learning only for the text modality during training, and due to their large parameter size, the use of BERT models was not considered.
Q: Part of Hypergraph Convolution very interesting. But how it will work in high-dimensional space? for example, when analyzing spectral characteristics?
A: The operation of hypergraph convolution in multimodal conversational emotion analysis mainly focuses on modeling the relationships in multimodal data to better capture the information interactions between different modalities, thereby improving the performance of emotion analysis. In high-dimensional space, after hypergraph convolution operations, the obtained node representations may contain information from different modalities. These pieces of information can be integrated into the final multimodal representation through operations such as pooling and fusion. Moreover, hypergraph convolution operations consider the relationships between nodes and hyperedges for information propagation. This operation can capture the mutual influences between modalities, enabling the model to better understand the comprehensive information in multimodal data.

Reviewer 4 Report
Comments and Suggestions for Authors
Authors presents an emotion recognition system which integrates audio, video and text data by means of hypergraphs. Results show significative improvements both over single modal and multimodal systems, mainly due to features like inter-modal connection modelling and a dynamic time window.
Author Response
Thank you very much for your review of my paper. I have revised the paper as required, and the following are the answers to the questions.
Authors presents an emotion recognition system which integrates audio, video and text data by means of hypergraphs. Results show significative improvements both over single modal and multimodal systems, mainly due to features like inter-modal connection modelling and a dynamic time window.
A: Thank you for the reviewer's feedback. I have refined the overall English writing of the paper and provided detailed descriptions for some of the methods. Thank you.
